# Gorlin Syndrome: Recent Advances in Genetic Testing and Molecular and Cellular Biological Research

**DOI:** 10.3390/ijms21207559

**Published:** 2020-10-13

**Authors:** Shoko Onodera, Yuriko Nakamura, Toshifumi Azuma

**Affiliations:** 1Department of Biochemistry, Tokyo Dental College, 2-9-18 Kandamisaki-cho Chiyoda-ku, Tokyo 101-0061, Japan; onoderashoko@tdc.ac.jp; 2Department of Oral Oncology, Oral and Maxillofacial Surgery, Tokyo Dental College, 5-11-13 Sugano, Ichikawa, Chiba 272-8513, Japan; nakamurayuriko@tdc.ac.jp

**Keywords:** Gorlin syndrome, nerved basal cell carcinoma, hedgehog pathway

## Abstract

Gorlin syndrome is a skeletal disorder caused by a gain of function mutation in Hedgehog (Hh) signaling. The Hh family comprises of many signaling mediators, which, through complex mechanisms, play several important roles in various stages of development. The Hh information pathway is essential for bone tissue development. It is also the major driver gene in the development of basal cell carcinoma and medulloblastoma. In this review, we first present the recent advances in Gorlin syndrome research, in particular, the signaling mediators of the Hh pathway and their functions at the genetic level. Then, we discuss the phenotypes of mutant mice and Hh signaling-related molecules in humans revealed by studies using induced pluripotent stem cells.

## 1. Introduction

Gorlin syndrome (GS) is a hereditary disease characterized by systemic and diverse developmental abnormalities and neoplastic lesions. While also known as nevoid basal cell carcinoma syndrome or basal cell naevus syndrome, this rare genetic disorder was first reported by RJ Gorlin [1]. The estimated prevalence of GS is 1/31,000 in the United States of America [2], 1/256,000 in Italy [3,4], 1/164,000 in Australia [5] and the United Kingdom [6,7] and 1/235,800 in Japan [8], with a male-to-female ratio of 1:1 [8]. GS is characterized by the development of multiple odontogenic keratocysts, which is frequently seen in young patients below the age of 20 years, and basal cell carcinomas (BCCs), which are usually observed in middle-aged patients (Figure 1). Rahbari and Mehregan have found that 2% of patients under the age of 45 with basal cell carcinoma have GS [9]. The incidence rates vary widely by ethnicity. Only about 40% of black patients affected by GS show BCC, while up to 90% of whites have been reported to show BCC. Approximately 5% of patients with GS develop medulloblastoma [10]. Compared to non-symptomatic medulloblastoma, medulloblastoma in GS occurs at an early age of 1–2 years [11,12]. The prevalence of medulloblastoma in GS is significantly higher in patients with the *SUFU* gene mutation (33%) than in patients with the patched 1 (*PTCH1*) gene mutation (<2%) [10,11,13]. Despite these severe major clinical manifestations, including multiple basal cell carcinomas (BCCs), medulloblastomas, life expectancy is not reduced in patients with GS. About 60% of individuals have macrocephaly, frontal ridges, coarse facial features, and facial milia. Additionally, most patients exhibit skeletal abnormalities such as bisecting ribs and wedge-shaped vertebrae. Heart fibroma is observed in approximately 2% of patients and ovarian fibroma is seen in approximately 20% of GS patients [10,11,13], odontogenic keratocysts (OKC) of the jaw, hyperkeratosis of the palms and soles (palmar and/or plantar pits), ectopic calcification, especially in the cerebral stem, or intracranial ectopic calcification of the falx is found in more than 90% of patients by the age of 20 [14,15,16,17,18]. Macrocephaly with frontal elevation, cleft lip or cleft palate, and eye abnormality are also seen in some patients. Intellectual disability is found in up to 5% of patients [16].

The genes responsible for GS are patched 1 (*PTCH1*), *PTCH2*, and *SUFU*. More than 100 gene mutations have been reported so far. In 1993, Evans and colleagues proposed clinical criteria for the diagnosis of GS based on the most frequent or specific features of the syndrome [6]. To date, different diagnostic criteria have been proposed in the United States [19,20], the United Kingdom [6], and Australia [5,20] for GS; however, the criteria reported by Kimonis et al. [19] (USA) are the most frequently used. This method includes six major (basal cell carcinoma, jaw bone cyst, palmar plantar small depression, calcified falx cerebri, rib abnormalities, and family history of the disease within the first degree) and six minor criteria (macrocephaly, congenital malformation, skeletal abnormalities, X-ray abnormalities, ovarian fibromas, and medulloblastomas) (Table 1). The clinical diagnosis of Kimonis requires fulfillment of either two major criteria, or one major and two minor criteria [19]. No study has been able to assess which combination of diagnostic criteria represents the best tradeoff between sensitivity and specificity. Furthermore, many symptoms other than those mentioned above have been observed in GS patients. The diagnosis is confirmed by gene mutation analysis, and genetic counseling is required after genetic diagnosis [20].

## 2. Genetic and Molecular Structural Aspects of Gorlin Syndrome

The most common mutation in this syndrome is the *PTCH1* gene mutation. The detection rates of *PTCH1* mutations were over 90% [21,22,23,24]. *PTCH1* is a 12-transmembrane receptor located on the long arm of chromosome 9 (9q22.32) and consists of 23 exons. It is a 12-transmembrane (TM) domain membrane protein with two extended ectodomains (ECD1 and ECD2) involved in ligand recognition, and a member of the RND transporter family (Figure 2). The TM2-6 bundle of *PTCH1* is annotated as a sterol sensing domain (SSD) based on the sequence similarity with the domains in proteins involved in cholesterol homeostasis. SSDs are believed to be involved in interactions with Hedgehog (Hh) ligands, as suggested by recent structural studies [25]. The structure of the ecto-luminal domain (ECD1, ECD2) and transmembrane domain of Ptch1 has been shown using cryo-EM. These structures revealed that the two ecto-luminal domains (ECD1 and ECD2) and the two SSD regions form distinct but closely spaced modules [24]. It also provides insights into the specific mutations found in Ptch1 in GS, namely those at the center of the interface between two SSD-like modules [26]. However, mutation clustering has not been confirmed. No association between the genetic variant and phenotype of this disease has been observed. Exons 2 to 21 were evenly distributed, and no clear hotspots were observed [27,28]. The most common form of mutation is frameshift mutation, followed by nonsense mutation, leading to the premature termination of *PTCH1* translation (Appendix A). Sanger sequence analysis could not identify large deletions in 6–21% patients. There are also reports of splicing mutations, large deletions, and missense mutations. For missense mutations, a combination of disease diagnosis in clinical conditions, segregation analysis, and in silico predictive programs for protein dysfunction should be carefully considered to determine if the mutation is responsible (Appendix A). To date, most GS diagnoses are based on clinical manifestations and some reported criteria, such as the Kimonis diagnostic criteria. In fact, the diagnostic criteria of Kimonis do not require genetic testing. Today, the progress of NGS (next generation sequence) technology is remarkable, and if a genetic diagnosis panel using NGS is developed in the future, genetic diagnosis will be easier, faster, and cheaper, and there is no doubt that it will be indispensable for GS diagnosis. A detailed correlation between GS mutation details and clinical symptoms may gradually reveal a correlation between the location of missense mutations and clinical symptoms. While 80% of sporadic keratocystic odontogenic tumor (KCOT) reports are due to *PTCH1* biallelic mutations [29], GS reports are not associated with such cases; instead, all are cases of heteroinsufficiency. 

As described above, the high incidence of biallelic somatic *PTCH1* mutation is quite high in KCOT (80% of the cases), and somatic *PTCH1* mutation seems to be quite characteristic of this disease. However, such somatic mutations have also been reported as mosaic mutations in GS [30,31]. Other rare causative genes such as *SUFU* and *PTCH2* genes have also been detected in GS. SUFU is involved in the suppression of the Hh signaling pathway, and it has been reported that this loss of function causes constitutive activation of the Hh signaling pathway and causes GS [32].

According to the previous report from Netherland, nine out of 171 individuals (5.2%) with GS had *SUFU* mutations. As described above, patients with *SUFU* mutations have a significantly increased risk of developing medulloblastoma [12]; however, no patient diagnosed as GS with *SUFU* mutations has been reported to develop KCOT.

*PTCH2*, a close relative of *PTCH1*, has also been reported as a rare causative gene for GS. There are only a couple of families of GS due to germline mutations in the *PTCH2* gene reported so far, neither of which includes a family of medulloblastomas. Patients with the *PTCH2* mutation are more likely to have a milder phenotype than those with *PTCH1* mutation-positive classical GS [33]. Recently, it was shown that, in GS, causative gene mutations may exist in both *PTCH1* and *PTCH2* simultaneously. Onodera et al. [34] detected mutations in *PTCH2, BOC*, and *WNT9b* genes along with *PTCH1* by exome sequence analysis. These mutations are predicted to have a functional impact by programs called MutationTaster and Polyphen2 (Polymorphism Phenotyping v2), a web-based program to evaluate DNA sequence variants for their disease-causing potential. These results indicate that some patients with GS might have mutilayered mutations in the Hh signaling pathway [34].

Next generation sequence analysis may pave the way for a new paradigm of GS genotyping. *PTCH1* mutations are often undetected in some patients because the routine Sanger sequence analysis is labor intensive and time consuming [33]. Mutations that occur outside the *PTCH1* analysis region, changes in copy number variation (CNV), or the existence of another causative gene also contribute to difficulty in the detection of GS. Morita et al. [35] reported that NGS analysis found specific single-nucleotide mutations (SNVs) in *PTCH1* and its surrounding *PTCH1* and copy number alterations (CNAs) in cases where the causative SNV was not detected. They established a custom HaloPlex panel containing genes involved in the Hh-related pathways *PTCH1*, *PTCH2*, *SHH*, *SUFU*, *SMO*, *GLI1*, *GLI2*, and *GLI3* [35]. Shiohama et al. investigated microRNA profiles in GS patient-derived fibroblasts [36]. They found down-regulation of hsa-miR-196a-5p and up-regulation of hsa-miR-4485. The hsa-miR-196a-5p complementarily binds to the 3’UTR site of MAP3K1 and is considered to be the target. While MAP3K1 expression is elevated at the mRNA and in protein levels in GS patient-derived fibroblasts, hsa-miR-196a-5p expression is reduced. Activation of the Hh pathway in a mouse mesenchymal cell line reduced miR-196a-5p expression and increased MAP3K1 expression. MAP3K1 has been reported to activate Hh signaling. Therefore, suppressing the expression of hsa-miR-196a-5p is considered to act as a positive feedback for the activation of the Hh pathway.

## 3. Genomic Instability

How does Hh activation in GS result in tumorigenesis? Genetic instability related to canonical Hh signaling is an important clue to understanding the pathogenesis of GS. Human genomic DNA is constantly under attack by both endogenous and exogenous DNA damaging factors [37]. It is estimated that every ~10^13^ cells in the human body encounter tens of thousands of DNA damage factors per day [38]. These genetic instabilities were closely related to the Hh signaling pathway. Indeed, mutations in the Hh signaling pathway lead to genetic instability and various cancers [39]. For example, the *PTCH1* germline mutation causes many BCCs in patients and, rarely, rhabdomyosarcoma (RMS). Constitutive activation of canonical Hh signaling in GS affects the expression of several oncogenes that trigger cell division and regulate cell cycle checkpoints and DNA repair mechanisms [33]. The other evidence is that Gli1 increases apoptosis and DNA damage. Ptch heterozygous mice (ptc^+/−)^ are known to develop increased spontaneous medulloblastoma [40]. It was demonstrated that elevated GLI1 in these mice abrogates ataxia telangiectasia and Rad3-related (ATR)- checkpoint kinase 1 (CHK1)-signaling, induces spontaneous and irradiation-induced genome instability, and promotes tumor development. ATR (ataxia telangiectasia and Rad3-related (ATR) proteins) are key regulators of the DNA damage response and maintain genome integrity in eukaryotic cells. ATR upregulates cell checkpoint kinase 1 and induces cell cycle arrest and DNA repair. Therefore, GLI1 upregulation inhibits DNA repair by upregulating ATR-CHK1 signaling; however, there are some conflicting observations have been reported with regard to Gli and genetic instability. Overexpression of *GLI1* blocks CHK1 activation, but depletion of GLI1 also causes the disruption of CHK1 activation. Both the overexpression and depletion of GLI1 can lead to genomic instability [41]. Inhibition of GLI1 induces DNA damage activation of ataxia telangiectasia-mutated protein kinase (ATM)-checkpoint kinase 2 (CHK2), and cell cycle arrest in the early S phase [42]. Thus, although constitutive activation of Hh signaling in GS must take part in genetic instability and tumorigenesis, more investigations are needed to reveal the precise mechanism.

## 4. Bone Metabolism, Hedgehog Over-Activation and Pathological Mechanism

To clarify the mechanism underlying the association between heterozygous *PTCH1* mutations/deletions and their associated Gorlin syndrome (GS)-related phenotypic manifestations, two independent lines of *Ptch1-*deficient mice were constructed: *Ptch1^+/−^* (exon 1/2) and *Ptch1^neo 67/+^* mice [40,43]. Both models were found to be prone to tumor development and skeletal abnormalities and exhibited increased susceptibility to irradiation [44].

Hedgehog-PTCH1 signaling plays a major role in osteoblast differentiation in endochondral bone formation and adult bone homeostasis. Indeed, mice lacking Indian hedgehogs (Ihh) cannot form osteoblasts or activate PTHrP expression in articular chondrocytes [45]. Ohba et al. [46] investigated bone metabolism in *Ptch1* +/*−* mice using von Kossa staining and calcein labeling. *Ptch1* +/− mice showed enhanced osteoclastogenesis and osteoblast differentiation, resulting in high bone mass in adult mice due to an increase in bone formation.

Furthermore, the expression of Gli3, one of three Gli family hedgehog (Hh) target transcription factors, is reduced in osteoblast precursor cells of Gorlin patient-derived cells [46]. Full-length Gli3 contains two active domains and one repression domain. Gli3 is cleaved by a PKA-dependent protease to produce a truncated repressor form with one active domain and one repression domain [47]. The conversion of full-length Gli3 (usually considered as pro-osteogenic) to repressed Gli3 was shown to be reduced not only in *Ptch1* +/− mice but also in GS patient-derived induced pluripotent stem (iPS) cells. This inhibitory Gli3 competes with Runx2, a master transcription factor for osteoblast differentiation, for DNA binding, suppresses Runx2 function, and reduces bone formation [46,48].

In sharp contrast, *PTCH1* gene knockout in mature osteoblasts increased osteoclastogenesis, leading to decreased bone mass and even osteopenia. Mak et al. [49] showed that the activation of Hh signaling increased bone formation, but the resulting bone is very fragile and porous due to severe bone resorption. In contrast, inhibition of Hh signaling in mature osteoblasts resulted in increased bone mass and protection from bone loss in older mice.

Hong et al. [50] identified germline *PTCH1* heterozygous mutations in all fibroblasts collected from patients with GS, and used these fibroblasts to identify differences in protein expression using tandem mass tag labeled proteomics analysis. They showed that SPARC (osteonectin) expression was significantly downregulated in GS_stromal cells compared with non-syndromic stromal cells. *PTCH1* siRNA transfection demonstrated that *SPARC* downregulation correlates with decreased *PTCH1* expression. Furthermore, exogenous SPARC promoted the osteogenic differentiation of stromal cells derived from *Ptch1* +/− mice with the enhanced development of calcium nodules. In addition, bone mineral density tests showed that patients with GS exhibit weak bone mass compared with sex- and age-matched controls. This study indicated that germline *PTCH1* heterozygous mutations play a major role in bone metabolism in patients with GS, particularly in those with PTCH1 protein truncation mutations. SPARC may represent an important downstream modulator of PTCH1 mediation of bone metabolism. Thus, bone mineral density monitoring is critical for patients with GS for the prevention of osteoporosis. Therefore, surgical procedures on syndromic keratocystic odontogenic tumors (KCOTs) should be performed only with due consideration of the weak bone mass in these patients.

As indicated above, some conflicting results have been reported in the literature. Ohba et al. observed that *Ptch1* +/− mice showed increased bone formation and cancellous bone mass [46]. However, Mak et al. found that mice with *Ptch1*-null mature osteoblasts showed increased osteoclast formation, which resulted in decreased bone mass [49]. Thus, it is not clear as to why conflicting results are observed in heterozygous vs. conditional KO mice. Many environmental factors, including epigenetic factors, can influence the bone phenotype and they require detailed study.

## 5. Tumors

### 5.1. Skin Cancer

Nevoid basal cell carcinoma, multiple nevoid basal cell epithelioma (basal cell carcinoma (BCC)) of the skin is caused by aberrant Hh signaling. Skin BCCs account for approximately 90% of all skin cancers [50,51,52,53,54,55]. The number is high in middle-aged and older people over 40 years old, and the number of outbreaks continue to increase as the population ages [56] Without treatment, BCC can invade not only the skin but also deep tissues, such as muscle and bone. However, metastasis to lymph nodes and internal organs is extremely rare, and the prognosis is not as bad as other types of malignant tumors [57]. The initial symptoms are the appearance of small black spots that are often mistaken for moles. Following this, they usually grow gradually over several years; the central part collapses, and the peripheral part is covered with black dyke-like artifacts (nodule/ulcer type). Most occur around the midface, around the nose, and upper lip. Rarely, some resemble scar-like forms of wounds, and some appear like eczema or rashes that do not look like cancer. There are usually no subjective symptoms, such as pain or itching [58].

It is known that major oncogene mutations occur first. In other words, it is thought that, when a tumor suppressor gene is inactivated or a protooncogene is activated, clonal growth occurs, and multiple gene mutations accumulate in duplicate, resulting in cancellation [59,60,61,62,63,64]. Inactivation of the *PTCH1* gene (73%) is known as inactivation of the tumor suppressor gene in BCC, and other cancer-related gene mutations that cause BCC carcinogenesis include *SMO* activation mutations and negative regulators of the Hh pathway. This is caused by loss-of-function mutations in the *SUFU* (8%) and *TP53* (61%) mutations [52,65]. Therefore, it is well accepted that BCC is primarily driven by the activation of the Hh pathway.

The BCC mutant gene is mapped to the human chromosome 9q22 and then to the *PTCH1* gene using gene linkage studies [66]. Mutations in *PTCH1* lead to constitutive activation of sonic Hh signaling, which promotes cell proliferation. However, strong epidemiological and genetic evidence has already indicated that UV-induced DNA damage is a major cause of patched1 inactivation [67,68,69,70]. Many sporadic—non-familial—BCC showed mutations in the PTCH allele that occurred at di-pyrimidinic sites, and were C to T transitions, and could, therefore, have been caused by UV radiation. Additional drivers for other cancer-related genes include *MYCN* (30%), *PPP6C* (15%), *STK19* (10%), *LATS1* (8%), *ERBB2* (4%), *PIK3CA* (2%), *NRAS KRAS*, or *HRAS*, all of which carry the mutation (2%) [71].

### 5.2. Medulloblastoma

Medulloblastoma is a brain tumor classified as a central nervous system embryonal tumor. It consists of undifferentiated small cells that develop in the posterior fossa, but there are various theories about the origin cells [71,72,73,74]. According to the WHO classification, there are five organizational types [75]: classic medulloblastoma, anaplastic medulloblastoma, large cell medulloblastoma, fibrogenic/nodular medulloblastoma, medulloblastoma, and extensive nodule formation. Approximately 5% of children with basal cell nevus syndrome develop medulloblastoma (undifferentiated neuroectodermal tumor: PNET) [39], which is often fibrogenic and medulloblastoma in GS [76]. The peak of onset is at the age of 2 and is characterized by onset at a young age [77].

Major gene mutations are frequently observed in *PTCH1*, *SMO*, and *SUFU*; however, *GLI2* and *MYCN* mutations have also been reported [78]. They are thought to arise from cerebellar granule cells [79]. In general, tumors with deficient mutations of TP53 have a very poor prognosis, as do medulloblastoma [80]. It has been suggested that Sox2+ or CD15+ tumor cells act as cancer stem cells. CD15 is a carbohydrate epitope expressed on normal neutrophils, Hodgkin lymphoma cells, and various solid tumor cells. CD15 is expressed on stem cells and tumor cells in various tissues. The fact that the presence of the CD15 gene expression signature identified tumor proliferating cells has led to its use as an important predictor of human medulloblastoma prognosis. The role of Sox2 involves both tumorigenesis and tumor maintenance; it is a stem cell driver, and Sox2 is now considered a cancer stem cell marker of medulloblastoma cells [81].

GS patients are clinically very sensitive to radiation, and these patients develop new BCC in the field after radiation therapy. In particular, it is known that BCC often occurs in the irradiated area after radiotherapy for medulloblastoma [6,14,82,83]. It is also responsible for the premature death of patients with GS. For general medulloblastoma, combination therapy involving surgical resection, chemotherapy, and radiation therapy is often followed, but, in this disease, medulloblastoma, BCC, osteochondral tumors, and nerve sheaths are consistent with the irradiation area. Therefore, it is necessary to be careful even when exposing the patients to radiation for diagnosis.

### 5.3. Keratocystic Odontogenic Tumor

KCOT is a major symptom of GS or as a sporadic lesion [84]. Multiple KCOTs are characteristic symptoms that indicate GS [85]. In particular, if multiple KCOTs are observed, it is necessary to suspect GS even if there are no other symptoms. Notably, keratocystic odontogenic tumors have a particularly high risk of recurrence [86]. Common symptoms of KCOT include pain, soft tissue swelling, bone dilatation, and paresthesia of the lips and teeth. Pathological fractures are seen when the KCOT expands beyond the medullary canal [87,88,89]. While Stoelinga et al. showed that epithelial islands accumulate on the surface of the excised keratinous cyst [90,91], many other studies have suggested that most of the epithelial islands on the wall of KCOT are attached to the mucosa covering the pathological tissue. The remaining epithelial residues may later cause cyst recurrence. Thus, basal epithelial derivatives may be involved in the etiology of KCOT [92], necessitating adequate curettage during surgical resection.

In 2005, the World Health Organization (WHO) changed the OKC histology from cysts to tumors. This reclassification was based on the fact that “invasive growth”, “post-treatment recurrence”, and “*PTCH1* mutations” are often found in OKC. However, in 2017, the WHO again reclassified OKCs as cysts because OKCs completely regress after decompression, and the intima of such decompressed cysts show histologically normal oral mucosa rather than OKCs.

Genetic mutations in the Hh receptor *PTCH1* and loss of heterozygosity (LOH) play important roles in the pathogenesis of KCOT [92]. Using a next-generation sequencing panel to identify recurrent genomic aberrations in sporadic KCOTs [93], Stojanov et al. [29] identified *PTCH1* inactivating mutations in 93% of cases, with biallelic inactivation in 80% of cases. Almost all mutations were found in the extracellular loop domain, which is important for SHH ligand binding and for the inhibitory effects on SMO. Noteworthily, unlike GS, in sporadic KCOT, no active mutations were observed other than in PTCH1 in the Hh pathway. Furthermore, in sporadic KCOT, somatic mutations of PTCH1 are often bialleic, indicating that the tumor suppressor effect of PTCH1 might be essential for KCOT development [93]. From a clinical perspective, vismodegib (Erivedge^®^), a US Food and Drug Administration (FDA)-approved targeted Hh pathway inhibitor, was reported to reduce KCOT in some patients with GS and may prove an alternative to surgical treatment [29]. These evidences indicate that Hh activation plays a role in KCOT pathogenesis.

### 5.4. Other Tumors

Cardiac and ovarian fibromas occur in approximately 2% and 20% of females, respectively [6,94]. Ovarian fibromas or cardiac fibromas are usually present at birth or occur soon after [39]. Cardiac fibromas can be asymptomatic or can cause arrhythmia or obstruction of cardiac flow. Rhabdomyomas may occur in the heart as well as in other sites [95].

## 6. Cancer Development and Hh Pathway Activation

The Hh signal transduction pathway is one of the main components of the tissue differentiation mechanism during embryonic development, but its existence usually disappears rapidly after birth. Abnormal Shh activation in adulthood leads to neoplastic growth, such as medulloblastoma, the most common childhood brain tumor, and BCC, the most common skin cancer. Thus, it is particularly interesting how the Hh cascade develops cancers [15].

Since the Hh pathway is a major cell cycle regulator, its abnormal activation leads to carcinogenesis. The Hh pathway is involved in two modes of cell cycle regulation. First, Ptch1 binds to the maturation-promoting factor (MPF) composed of cyclin B1 and cyclin-dependent kinase 1 (CDK1) in the absence of ligand binding, and its activity is inhibited [96,97]. When Hh binds to Ptch1, cyclin B1 is released and cell cycle proceeds. The other mechanism of the Hh pathway leads to cyclin D and cyclin E transcription and promotes cell cycle progression [98]. Hh proteins are often synthesized by locally defined epithelial cells as precursors that are activated by autocatalytic cleavage of the C-terminal intein domain, which is self-splicing polypeptide with an ability to excise themselves from flanking host protein regions, and Hh ligands bind to transmembrane receptors patched 1 (*PTCH1*) or patched 2 (*PTCH2*) and stimulate the Hh pathway [39,99,100]. Vertebrate Hh signaling is transduced by primary cilia, which protrude from the surface of most cells. In the absence of the Hh ligand, PTCH1 localizes at the primary cilium and prevents SMO from entering the cilium by an unknown mechanism [101]. In the presence of a ligand, PTCH1 leaves the cilia, allowing SMO to accumulate in the primary ciliary membrane [102]. In cilia, SMO inhibits GLI3 repressor formation and activates GLI2 [103]. GLI2 enters the nucleus and promotes the transcription of Hh target genes. Mutant inactivating mutations (loss of function) in *PTCH1, PTCH2*, or *SUFU*, the activation of mutations in SMO (gain of function) or the amplification of GLI2 lead to the random activation of the expression of target genes in Hh, causing cancer [104]. Cancers with aberrant activation of the Hh signaling pathway include BCC, medulloblastoma, pancreas, breast, colon, ovary, and small-cell lung cancer [105].

Until recently, there has been no approved therapy for advanced BCC. Vismodegib (Erivedge^®^) is a breakthrough drug developed for the treatment of advanced BCC [106]. The US Food and Drug Administration (FDA) approval for vismodegib (Erivedge^®^) was obtained in January 2012. The European Medicines Agency (EMA) granted approval to vismodegib (Erivedge^®^) for the treatment of adult patients with symptomatic metastatic BCC or locally advanced BCC inappropriate for surgery or radiotherapy on July 2013. Currently, vismodegib and sonidegib are considered promising cancer treatments for patients with refractory/advanced cancer. The Hh inhibitor bismodegib is approved for treatment of advanced BCC and metastatic BCC. Sonidegib has been approved for treatment of advanced BCC in the United States and Europe on 2013, and is approved in Switzerland and Australia for treatment of advanced and metastatic BCC on 2016 [33,107,108]. Radiation therapy for medulloblastoma in GS patients has been reported to induce invasive BCC [27]. Therefore, although radiation therapy is used for sporadic BCCs, GS patients should avoid radiation therapy for BCC, if possible. Vismodegib is useful when radiation therapy is contraindicated or when the lesion persists [109]. Vismodegib reduces the burden of BCC tumors and prevents the growth of new BCCs in patients with GS [106]. The low incidence of serious adverse effects and the efficacy of the drug on BCC suggest that vismodegib may be suitable for GS patients [110]. However, more than half of GS patients who received vismodegib treatment were discontinued due to treatment-related adverse events such as muscle cramps, hair loss, weight loss, and fatigue [111]. Discontinuation of vismodegib showed “rapid BCC recovery in GS patients”. Unlike medulloblastoma, there are no reports of treatment resistance in BCC [33]. As noted in the following section, Smo antagonists such as vismodegib reduce the burden on KCOT as well as BCC, and surgical resection is less cumbersome [106]. However, problems such as side effects and drug resistance have been reported [110]. SMO antagonists activate the Ca^2+^-AMPK signal [65,112], which is involved in the type II non-canonical Hh signal [65], and they are believed to affect energy metabolism in muscle, adipose tissue, the liver, and the pancreas, leading to weight loss and muscle side effects in SMO antagonists, such as convulsions, alopecia, fatigue, and dysgeusia, occurring in 19–33% of patients with a mortality rate of 2–11% [113].

## 7. Disease-Specific Induced Pluripotent Stem Cells

In 2006, Takahashi and Yamanaka succeeded in reprogramming mouse fibroblasts to generate induced pluripotent stem cells (iPS cells) for the first time. In the following year, they succeeded in establishing human iPSCs [114]. These iPSCs were established from mouse or human fibroblasts by the forced expression of four major transcription factors (Oct4, Sox2, Klf4, and c-Myc). To date, several researchers have shown various methods of producing iPS cells, and not only fibroblasts, but also cells contained in peripheral blood or even urine can be used [115,116]. In order to avoid the risk of malignant transformation of iPS cells as much as possible, the transcription factor induction method uses Sendai virus and episomal vectors, which have a low risk of incorporating the vector sequence into the original gene [115]. Human induced pluripotent stem cells (hiPSCs) are ideal regenerative medicine tools that have the same pluripotency as embryonic stem cells while reprogramming somatic cells to retain their genetic information. Patient-derived hiPSCs are also used for in vitro disease modeling and drug screening. Recently, iPSCs derived from patients with GS have been reported by several groups [31,48,117,118,119,120]. By using iPSCs derived from patients with GS, a model of abnormal tissue development and tumor formation can be constructed. This may lead to the elucidation of disease mechanisms and the development of therapeutic methods.

Several hiPSC-derived studies focusing on identifying DNA mutations have also been used to identify somatic mutations. Ikemoto et al. showed that genetic analysis of iPSC clones can contribute to the detection of mosaicism involving a minority population carrying a second mutation [31].

Aberrations in SHH signaling, especially *PTCH1* gene mutations leading to constitutively activated Smo, are found in ~25% of medulloblastoma. Similarly, approximately 15% of *Ptch1* +/− transgenic mice develop medulloblastomas [121]. Neuroepithelial stem (NES) cells, isolated from neural rosettes derived from human pluripotent stem cells, can grow for a long time in culture, and can be cultured into cerebellar granule neural precursor (GNP) cells. These GNP cells are thought to be the precursors of medulloblastoma, a common malignant brain tumor in children and young adults [76]. In fact, Huang et al. demonstrated that orthotopic transplantation of NES cells derived from a patient with GS generated medulloblastoma [120]. Both mutations in *DDX3X* and the loss of GSE1 accelerated tumorigenesis in Gorlin [120]. Ikemoto U et al. [118] showed iPSCs derived from fibroblasts of four patients with GS with heterozygous mutations of the *PTCH1* gene developed into medulloblastoma in 100% (four out of four) of teratomas generated by transplantation of iPS cells into immunodeficient mice. One of the medulloblastomas showed a loss of heterozygosity in the *PTCH1* gene, indicating a close clinical correlation with *PTCH1* heterozygosity and medulloblastomas. Ikemoto et al. discovered two adjacent gene mutations in the Gorlin iPSC. The iPS cell clone derived from a single somatic cell inherits the original somatic mutation as it is. These facts indicate that somatic mosaicism of *PTCH1* was observed in patients with Gorin syndrome. Thus, there is a common germ cell mutation in the *PTCH1* gene and different somatic cell mutations in each iPS cell clone derived from the same individual. Therefore, the iPS technique could contribute to the detection of the mosaic phenomenon existing in the *PTCH1* gene [31].

Hh signaling is essential for osteoblast differentiation. As described above, some previous reports have indicated its importance in bone homeostasis. However, even though studies on transgenic mice indicated the pathological significance of Hh signaling, the human disease model provides not only pathological clues, but also clinical applications, such as drug development.

Hasegawa et al. [119] from our group showed that iPS cells specific to patients with GS responded more sensitively to osteostimulation. We showed that basic FGF, IGF-1, and TGF-β stimulation or Smo agonist SAG treatment significantly upregulated osteoblast markers and ALP activity. BMP2/7 stimulation, a potent bone-induced growth factor, increased RUNX2 expression compared to normal iPS cells. GLI1 and GLI2 transcription factors decreased, while the repressor type GLI3 levels increased [119]. These findings indicate that Gorlin iPS cells are more prone to osteoblast differentiation than normal iPS cells.

## 8. Conclusions

The Hh information pathway is essential for many tissue developments and is crucial for the initiation of bone tissue development. It is also the major driver gene in the development of BCC and medulloblastoma. Therefore, the Hh pathway plays a major role in the pathogenesis of Gorlin syndrome. However, due to the complexity of the regulatory mechanism, there are several issues that need to be resolved. This review introduces some of them. In recent years, many studies have reported using iPS cells unique to Gorlin syndrome for the development of modern therapeutics and the results are highly promising.

## Figures and Tables

**Figure 1 ijms-21-07559-f001:**
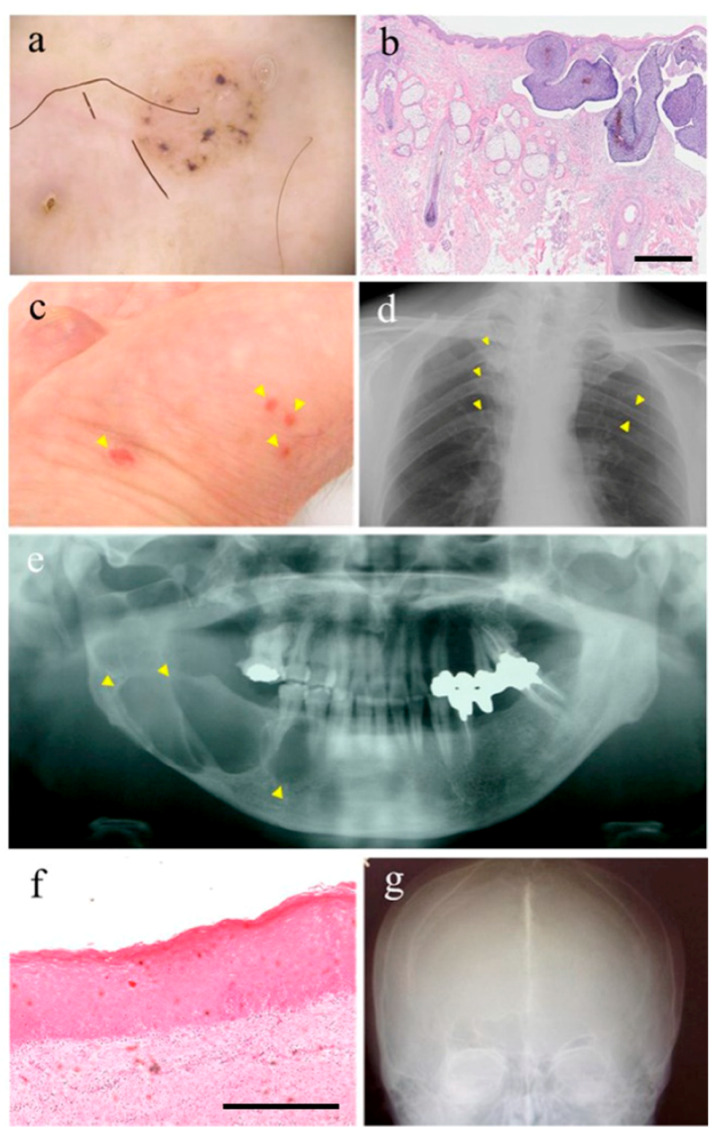
Characteristic features of Gorlin syndrome: (**a**) dermoscopic features of Basal cell carcinoma (BCC); (**b**) pathological phenotype of BCC. Scale bar: 50 μm; (**c**) multiple palmar pits are indicated by yellow arrowheads; (**d**) rib anomalies such as bifidity and splaying are indicated by yellow arrowheads; (**e**) keratocystic odontogenic tumor are indicated by yellow arrowheads (OKC); (**f**) pathological phenotype of OKC. Scale bar: 50 μm; (**g**) falx calcification.

**Figure 2 ijms-21-07559-f002:**
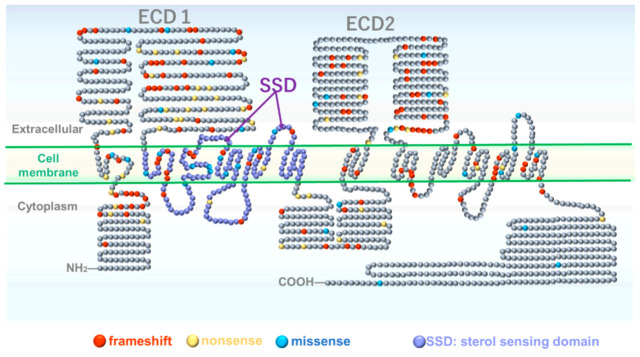
Patched 1 (PTCH1) gene mutation site and domain structure of Patched 1 protein. Each color dots showed the mutations in listed Appendix A. ECD1: extracellular domain 1, ECD2: extracellular domain 2, SSD: sterol sensing domain. Each dot represents one amino acid that consist PTCH1. Gray dots indicate amino acids for which no mutations have been reported.

**Table 1 ijms-21-07559-t001:** Gorlin syndrome diagnostic criteria.

**Major Criteria (Diagnostic Criteria for GS—from Kimonis, V.E. et al. [19])**
More than 2 BCCs or one under the age of 20 years. Odontogenic keratocysts of the jaw proven by histology. Three or more palmar or plantar pit. Bilamellar calcification of the falx cerebri. Bifid, fused or markedly splayed ribs. First degree relative with GS syndrome.
**Minor Criteria (Any One of the Following Features)**
Macrocephaly determined after adjustment for height. Congenital malformations: cleft lip or palate, frontal bossing, “coarse face” moderate or severe hypertelorism. Other skeletal abnormalities: Sprengel deformity, marked pectus deformity, marked syndactyly of the digits. Radiological abnormalities: bridging of the sella turcica, vertebral anomalies such as hemivertebrae, fusion or elongation of the vertebral bodies, modeling defects of the hands and feet, or flame shaped lucencies of the hands or feet. Ovarian fibroma. Medulloblastoma.

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
