# Peer review of "Gorlin Syndrome: Recent Advances in Genetic Testing and Molecular and Cellular Biological Research"

_ijms, 2020, doi:10.3390/ijms21207559_

Round 1
Reviewer 1 Report
Gorlin syndrome
This paper describe Gorlin syndrome and several observations in regard to Hedgehog signaling. It is a nice topic and al large number of useful information. However, I have several comments.
There is a large number of repetitions. Please collect paragraphs describing the same or the same topic in one paragraph if possible. Do not use words/genes without a short introduction explanation.
Introduction.
Introduction to Gorlin syndrome starts two times, line 24 and line 42. I would prefer that the two paragraphs are collected into a single paragraph. Furthermore I would prefer that the paragraph starts with the most frequent symptoms. In line 30 -line 32 BSS is described which seems rather rare ? (2%??) . whereas in line 36 symptoms present in about 60% of the patients are described.
Furthermore I would also prefer that You start with presentation of the genes (in the present version line 54 ) especially because the genes are mentioned in line 35-36. And also line 250 is about genetics collect avoid repetition
Line 54 says that more than 100 gene mutations are reported, and a list is shown with mutations in PTCH1. I prefer that inclusion of how frequent mutations are in the genes involved PTCH1, PTCH2 and SUFU. Or if you only want to describe PTCH1 specific then make it more clear. A long list with PTCH1 mutations is in my opinion not very useful . If You have identified the unpublished ones. I think you should only describe these as the rest is published. If You disagree a possibility could be to include the list as supplementary
By looking into the HGMD database You will find a large number of mutations in PTCH1 (more than 500)
and 11 in PTCH2 and 54 in SUFU
- Bone metabolism
This is a nice chapter. I just do not get. The expression of gli3 is reduced in osteoblast precursor cells (98) Is this in NORMAL cells (and not GS). So the conclusion from (98, 100) is this that the repressor GLI3 is reduced in GS and therefore Runx2 and bone formation is increased?? Please clarify the text. According to the tittle 98 also increased bone mass is observed ?? You did not mention bone mass here. I think You should also mention bone bass because You in line 90 starts with IN SHARP contrast… (did the sharp contrast not referred to the BONE MASS? (because it seems, as far as I understood the bone formation is increased in both experiments, whereas a increase in bone mass is observed in the first ref (Ohba) by not in the second (Mak) …or in the third Hong.
Line 105 SPARC, include an introduction to SPARC
Line 111..cancelllous?? should it mean increased?
Line 124- The initial symptoms were appearance …are appearance (old results present)
Line 133 loss of allele on chromosome 9q22 is common ref 117. According to ref 117 deletion of 9q22 is observed in few cases. The deletion include PTCH1…rewrite this chapter.
Line 139-144 patch explain more
BCC and medulloblastoma and connection to Hh is repeated several times. Collect and avoid repetition.
Line 157 TP53 sox2+ CD15 + explain more
Line 214 C-terminal intein domain explain more
Line 283 other rare causative genes have also been detected. Does that mean Few uutation in other genes?
Line 299 (CAN) I think it is normally called copy number variation (CNV)
Line 324 ATR-CHK1 IR explain
Line 377 ALP, BMP2/7 explain
Line 285 patients with SUFU …(12) repetition
Line 357 -358 repetition mentioned earlier
Line 272 -274. Is evaluation of missense mutations connected to NGS? Or did you men that funtionel investigation of missense mutations to predict pathogenicity is necessary??
Author Response
Here, I submit my point-ny-point response. I submit word file as well. Thank you
To reviewer1
This paper describe Gorlin syndrome and several observations in regard to Hedgehog signaling. It is a nice topic and al large number of useful information. However, I have several comments.
There is a large number of repetitions. Please collect paragraphs describing the same or the same topic in one paragraph if possible. Do not use words/genes without a short introduction explanation.
Response
Thank you very much for your review. I made the best of my corrections by fully referring to your suggestions.
Point-by-point responses
# Introduction.
Introduction to Gorlin syndrome starts two times, line 24 and line 42. I would prefer that the two paragraphs are collected into a single paragraph. Furthermore I would prefer that the paragraph starts with the most frequent symptoms. In line 30 -line 32 BSS is described which seems rather rare ? (2%??) . whereas in line 36 symptoms present in about 60% of the patients are described.
Response
Thank you for your comment. I agree with your comment. I have combined two paragraphs into one paragraph. The reason why I started with medulloblastoma and BSS is that those are more life threatening symptoms than other symptoms.
BCC incidence rate under 40 y/o is about 2%, however, the incidence rates vary widely by ethnicity. Only about 40% of black patients affected by GS show BCC, but up to 90% of whites have been reported. Added a description of this part as follows.
LINE 33
The incidence rates vary widely by ethnicity. Only about 40% of black patients affected by GS show BCC, but up to 90% of whites have been reported.
# Furthermore I would also prefer that You start with presentation of the genes (in the present version line 54 ) especially because the genes are mentioned in line 35-36. And also line 250 is about genetics collect avoid repetition
Response
Thank you for your advice. Following the reviewer’s advice, I started with the presentation of the genes.
# Line 54 says that more than 100 gene mutations are reported, and a list is shown with mutations in PTCH1. I prefer that inclusion of how frequent mutations are in the genes involved PTCH1, PTCH2 and SUFU. Or if you only want to describe PTCH1 specific then make it more clear. A long list with PTCH1 mutations is in my opinion not very useful . If You have identified the unpublished ones. I think you should only describe these as the rest is published. If You disagree a possibility could be o include the list as supplementary
Response
Thank you for making a very important point. Following the reviewer's advice, the list of reported PTCH1 mutations observed in patients with Gorlin syndrome has been moved to supplemental data. Added the SUFU mutation rate found in Gorlin syndrome. However, PTCH2 is very rare and only a few reports have been published, so the incidence has not yet been calculated.
# By looking into the HGMD database You will find a large number of mutations in PTCH1 (more than 500)and 11 in PTCH2 and 54 in SUFU
I appreciate this comment. Those mutations were found not only in Gorlin but also in other diseases such as sporadic BCC.
# Bone metabolism
This is a nice chapter. I just do not get. The expression of gli3 is reduced in osteoblast precursor cells (98) Is this in NORMAL cells (and not GS). So the conclusion from (98, 100) is this that the repressor GLI3 is reduced in GS and therefore Runx2 and bone formation is increased?? Please clarify the text. According to the tittle 98 also increased bone mass is observed ?? You did not mention bone mass here. I think You should also mention bone bass because You in line 90 starts with IN SHARP contrast… (did the sharp contrast not referred to the BONE MASS? (because it seems, as far as I understood the bone formation is increased in both experiments, whereas a increase in bone mass is observed in the first ref (Ohba) by not in the second (Mak) …or in the third Hong.
Response
Line 174
Thank you for your important suggestion. I corrected the description as follows.
Ptch1 +/- mice showed enhanced osteoclastogenesis and osteoblast differentiation, resulted in high bone mass in adult mice due to an increase in bone formation.
# Line 105 SPARC, include an introduction to SPARC
Response
SPARC is osteonectin, well known integrin binding protein in bone tissues and I wonder if this word needs to be explained.
# Line 111..cancelllous?? should it mean increased?
Response
The cancellous bone mass exists inside the bone surrounded by the cortical bone mass, and Ohba et al. reported that this part was increased.
# Line 124- The initial symptoms were appearance …are appearance (old results present)
Response
Thank you for pointing out this mistake, I corrected as follows
Line 219
The initial symptoms are appearance of small black spots
# Line 133 loss of allele on chromosome 9q22 is common ref 117. According to ref 117 deletion of 9q22 is observed in few cases. The deletion include PTCH1…rewrite this chapter.
Response
Thank you for your comment. I have corrected this part as follows.
Line 227
BCC mutant gene is mapped to the human chromosome 9q22 and then to the PTCH1 gene using gene linkage studies [65]. This occurs by PTCH1 gene inactivation (73%) or SMO activating mutations in this region, and loss-of-function mutations in the negative regulator of the Hh pathway SUFU (8%) and TP53 61% mutations [52,66].
I deleted the following sentence.
Therefore, it is well accepted that BCC is primarily driven by the activation of the Hh pathway.
# Line 139-144 patch explain more
Response
A patch is a protein patched 1 encoded by the PTCH1 gene. I added following sentences.
Line231-237
Mutations in PTCH1 lead to constitutive activation of sonic Hh signaling, which promotes cell proliferation. However, strong epidemiological and genetic evidence has already indicated that UV-induced DNA damage is a major cause of patched1 inactivation [67–70]. Many sporadic – non-familial – BCC showed mutations in the PTCH allele occurred at di-pyrimidinic sites and were C to T transitions, and could, therefore, have been caused by UV radiation Many sporadic – non-familial – BCC showed mutations in the PTCH allele occurred at di-pyrimidinic sites and were C to T transitions, and could, therefore, have been caused by UV radiation.
# BCC and medulloblastoma and connection to Hh is repeated several times. Collect and avoid repetition.
Response
Thank you for your advice. I deleted several repeated parts.
# Line 157 TP53 sox2+ CD15 + explain more
Response
I added several explanations about sox2 and CD15 as follows.
Line 254
It is thought to arise from cerebellar granule cells [76]. In general, tumors wit mutations of TP53 have a very poor prognosis and as do medulloblastoma [77]. It has been suggested that Sox2+ or CD15+ tumor cells act as cancer stem cells. CD15 is a carbohydrate epitope expressed on normal neutrophils, Hodgkin lymphoma cells, and various solid tumor cells. CD15 is expressed on stem cells and tumor cells in various tissues. The fact that the presence of the CD15 gene expression signature identified tumor proliferating cells has led to its use as an important predictor of human medulloblastoma prognosis. The role of Sox2 is both tumorigenesis and tumor maintenance, and is a stem cell driver, and Sox2 is now considered a cancer stem cell marker of medulloblastoma cells [78].
# Line 214 C-terminal intein domain explain more
Response
Inteins are self-splicing polypeptides with an ability to excise themselves from flanking host protein regions. Inteins are widely dispersed in nature. Inteins are somehow analogous to introns. The splicing of inteins can occur either spontaneously or under favorable conditions.
Therefore inteins are not specific domain in Hedgehog thus I wonder if I need to add some descriptionin about intein in manuscript.
I added following descriptions
Line 317
which is self-splicing polypeptide with an ability to excise themselves from flanking host protein regions, ns as follow
# Line 283 other rare causative genes have also been detected. Does that mean Few uutation in other genes?
Response
Thank you for raising this important issue. It is well accepted that PTCH2 and SUFU has been recognized as causative gene for Gorlin syndrome.
# Line 299 (CAN) I think it is normally called copy number variation (CNV)
Response
Thank you. I used CNV instead of CAN as the reviewer recommended.
Line 125
copy number variation (CNV)
# Line 324 ATR-CHK1 IR explain
Thank you for your advice. I added following description to explain ATR-HK1
Response
Line 151
ATR (ataxia telangiectasia and Rad3-related (ATR) proteins) are key regulators of the DNA damage response and maintain genome integrity in eukaryotic cells. ATR upregulates cell checkpoint kinase 1 and induces cell cycle arrest and DNA repair. Therefore GLI1 upregulation inhibits DNA repair by upregulating ATR-CHK1 signaling, however, there are some conflicting observations have been reported with regard to Gli and genetic instability.
# Line 377 ALP, BMP2/7 explain
Response
Thank you for your advice.
I explained ALP as follows.
Line400
osteoblast marker ALP activity
I explained BMP2/7 as shown below.
Line 401
a potent bone-induced growth factor,
# Line 285 patients with SUFU …(12) repetition
Response
Thank you for making pointing out this repetition. I deleted this part.
Line 105- deleted.
# Line 357 -358 repetition mentioned earlier
Response
Thank you for making pointing out this repetition. I deleted this part.
# Line 272 -274. Is evaluation of missense mutations connected to NGS? Or did you men that funtionel investigation of missense mutations to predict pathogenicity is necessary??
Thank you for raising this important issue. I agree with the reviewers' comments. Below is a text with a detailed explanation.
Line91-97
To date, most GS diagnoses are based on clinical manifestations and some reported criteria, such as the Kimonis diagnostic criteria. In fact, the diagnostic criteria for Kimonis do not require genetic testing. Today, the progress of NGS technology is remarkable, and if a genetic diagnosis panel using NGS is developed in the future, genetic diagnosis will be easier, faster, and cheaper, and there is no doubt that it will be indispensable for GS diagnosis. A detailed correlation between GS mutation details and clinical symptoms may gradually reveal a correlation between the location of missense mutations and clinical symptoms.
Reviewer 2 Report
The review is well written but I would have some minor suggestions
Figure 1 not adequate on quality of the images . Please change.
Table 2 : too long :including all the list of mutations is not helpful. Please change the table and list the most common ones
Page 13 : paragraph "Until recently, there has been no approved therapy for advanced BCC. Vismodegib (Erivedge® )is a breakthrough drug developed for the treatment of advanced BCC [159]. The US Food and DrugAdministration (FDA) approval for vismodegib was obtained in January 2012. Currently, vismodegib and sonidegib are considered promising cancer treatments for patients with refractory/advanced cancer. The Hh inhibitor bismodegib is approved for treatment of advanced BCC and metastatic BCC."
Please add EMA approvement date for sonidegib.
Correct drug name " Sonicideb" has been approved for treatment of advanced BCC in the United States and Europe, and isapproved in Switzerland and Australia for treatment of advanced and metastatic BCC "
Suggest to refer these papers Basal Cell Carcinoma: A Comprehensive Review. Int J Mol Sci. 2020 Aug 4;21(15):5572.High-Risk Recurrence Basal Cell Carcinoma: Focus on Hedgehog Pathway Inhibitors and Review of the Literature.Chemotherapy. 2020 Aug 10:1-9.
Author Response
To Reviewer2
# The review is well written but I would have some minor suggestions.
Response
I greatly appreciate this positive comment..
# Figure 1 not adequate on quality of the images . Please change.
Response
I did our best to improve picture quality.
Table 2 : too long :including all the list of mutations is not helpful. Please change the table and list the most common ones.
Response
Thank you for the advice. I moved this table to supplementary data.
# Page 13 : paragraph "Until recently, there has been no approved therapy for advanced BCC. Vismodegib (Erivedge®) is a breakthrough drug developed for the treatment of advanced BCC [159]. The US Food and DrugAdministration (FDA) approval for vismodegib was obtained in January 2012. Currently, vismodegib and sonidegib are considered promising cancer treatments for patients with refractory/advanced cancer. The Hh inhibitor bismodegib is approved for treatment of advanced BCC and metastatic BCC."
Please add EMA approvement date for sonidegib (Erivedge®)
Response
Thank you for pointing out this important issue. European Medicines Agency (EMA) granted approval to vismodegib (Erivedge®) for the treatment of adult patients with symptomatic metastatic BCC or locally advanced BCC inappropriate for surgery or radiotherapy on July 2013. Sonidegib (Odomzo®) for the treatment of adult patients with locally advanced BCC on 2015. Description in the manuscript was corrected as follows.
Line 319 : Until recently, there has been no approved therapy for advanced BCC. Vismodegib (Erivedge®) is a breakthrough drug developed for the treatment of advanced BCC [103]. The US Food and Drug Administration (FDA) approval for vismodegib (Erivedge®) was obtained in January 2012. Eupopean Medicines Agency (EMA) granted approval to vismodegib (Erivedge®) for the treatment of adult patients with symptomatic metastatic BCC or locally advanced BCC inappropriate for surgery or radiotherapy on July 2013. Currently, vismodegib and sonidegib are considered promising cancer treatments for patients with refractory/advanced cancer. The Hh inhibitor bismodegib is approved for treatment of advanced BCC and metastatic BCC. Sonidegib has been approved for treatment of advanced BCC in the United States and Europe on 2013, and is approved in Switzerland and Australia for treatment of advanced and metastatic BCC on 2016 [33,104,105].
# Correct drug name " Sonicideb" has been approved for treatment of advanced BCC in the United States and Europe, and isapproved in Switzerland and Australia for treatment of advanced and metastatic BCC "
Response
Thank you for pointing this mistake. I wrote correct name ‘Sonidegib’ as shown above.
Suggest to refer these papers Basal Cell Carcinoma: A Comprehensive Review.
Dika E, Scarfì F, Ferracin M, et al.Int J Mol Sci. 2020 Aug 4;21(15):5572.High-Risk Recurrence Basal Cell Carcinoma: Focus on Hedgehog Pathway Inhibitors and Review of the Literature.Campione E, Di Prete M, Lozzi F, et al.Chemotherapy. 2020 Aug 10:1-9.
Round 2
Reviewer 1 Report
I think the paper is improved now.
However I think something get wrong with the section from line 226 to line 239, please rewrite. Note that PTCH1 is located at chromosome 9, but SMO is on chromosome 7, SUFO on chromosome 10, PTCH2 on chromosome 1 , TP53 chromosome 17. So "THIS REGION" is misleading.
Author Response
We appreciate and agree with the reviewer's instructive suggestion. Indeed, "THIS REGION" is misleading. Therefore, we deleted this part and re-write Line226-229 as follows.
Inactivation of the PTCH1 gene (73%) is known as inactivation of the tumor suppressor gene in BCC, and other cancer-related gene mutations that cause BCC carcinogenesis include SMO activation mutations and negative regulators of the Hh pathway. included. This is caused by loss-of-function mutations in the SUFU (8%) and TP53 (61%) mutations [52,65].